# Human Papillomavirus Self-Sampling for Unscreened Women Aged 24 Years During the COVID-19 Pandemic

**DOI:** 10.3390/healthcare12212160

**Published:** 2024-10-30

**Authors:** Yoko Chino, Toshimichi Onuma, Taro Ito, Akiko Shinagawa, Tetsuji Kurokawa, Makoto Orisaka, Yoshio Yoshida

**Affiliations:** 1Department of Obstetrics and Gynecology, Faculty of Medical Sciences, University of Fukui, Fukui 910-1193, Japan; yoyoyo@u-fukui.ac.jp (Y.C.); toonuma@u-fukui.ac.jp (T.O.); taroito@u-fukui.ac.jp (T.I.); sngw@u-fukui.ac.jp (A.S.); orisaka@u-fukui.ac.jp (M.O.); 2Department of Obstetrics and Gynecology, Tannan Regional Medical Center, Fukui 916-8515, Japan; 3Department of Obstetrics and Gynecology, Red Cross Fukui Hospital, Fukui 918-8501, Japan; 4Department of Obstetrics and Gynecology, Fukui-Ken Saiseikai Hospital, Fukui 918-8503, Japan; kurotetu@u-fukui.ac.jp

**Keywords:** cervical cancer, screening, human papillomavirus self-sampling, unscreened women, PCR-based HPV test, opt in

## Abstract

Background: The increasing trend of cervical cancer in women in their 20s in Japan is largely attributable to the low rate of cervical cancer screening. This study aimed to assess the usefulness of human papillomavirus (HPV) self-sampling among 24-year-old Japanese women who had never previously been screened for cervical cancer during the coronavirus disease (COVID-19) pandemic. Methods: In August 2021, consenting eligible women received HPV self-sampling kits. An Evalyn brush was used for self-sampling, and a Cobas 4800 PCR-based HPV DNA test was used to detect high-risk HPV genotypes. We analyzed the return rates of self-sampling kits and conducted a survey on the acceptability of the self-sampling method. Results: Of the total 1997 eligible women, 13.4% (268/1997) agreed to participate. The return rate of the kits was 72.4% (194/268), corresponding to 9.7% of the eligible population. Among the participants who returned the kits, 14.9% (29/194) tested positive for HPV, and 41.4% (12/29) of these underwent subsequent cytological testing. The questionnaire results indicated that 57.8% of participants reported no pain during self-sampling, and 72.9% expressed a willingness to continue using the self-sampling method in the future. Conclusion: This study demonstrated that opt-in HPV self-sampling among 24-year-old women who had never been screened for cervical cancer had a favorable kit return rate and was well accepted by the participants, especially during the COVID-19 pandemic. However, the follow-up cytology test rates were low, highlighting the need for improved post-screening management.

## 1. Introduction

Women in Japan in their 20s have a high incidence of cervical cancer. In 2013, the incidence rate was 6.20 per 100,000 population, as reported by the Global Cancer Observatory. In comparison, the United States and Australia reported cervical cancer incidence rates of 3.18 and 5.59 per 100,000 population, respectively [1]. In 2022, the incidence among Japanese women in their twenties had increased to 9.90 per 100,000 population, whereas the incidence rates in the United States and Australia were 4.31 and 3.17 per 100,000 population, respectively [1]. These findings underscore the high and escalating incidence and mortality rates of cervical cancer in Japan, contrasting with the trends observed in the majority of developed countries [2]. The significant factors contributing to this trend are the low HPV vaccination rate and the low acceptance of cervical cancer screening [2]. Despite the resumption of government recommendations, HPV vaccination coverage remains low, with only 2.83% of the vaccine-resumed generation (birth fiscal year 2010) receiving the vaccine [3]. For this reason, it is essential to increase the cervical cancer screening rate.

In Japan, cervical cancer screening is performed using cytological diagnosis, as mandated by a directive from the Director-General of the Health Bureau of the Ministry of Health, Labor and Welfare. Women aged 20 years or older are advised to undergo cytological screening for cervical cancer every two years [4]. However, the screening rate of eligible women in Japan is only 42.1%. Women are less likely to get screened for cervical cancer in Japan than in the United States or the United Kingdom (84.5% or 78.1%, respectively) [5]. The 2022 Comprehensive Survey of Living Conditions in Japan reported that only 13.6% of individuals aged 20–29 years had undergone cervical cancer screening within the 2 years [6].

The cervical cancer screening should be emphasized to women aged 20–24 years to reduce cervical cancer mortality and incidence in their 20s. In this age group, one of the most common reasons for nonparticipation in cervical cancer screening is the feeling of being too young [7]. Women in this age group may have limited experience with serious illnesses and are likely to be overly confident about their health. Furthermore, the lack of information on cervical cancer screening and its prevention has reduced this group’s willingness to seek medical attention [7]. However, those who have never been screened for cervical cancer are most likely to develop advanced cancer [8]. Regular checkups are important, especially to ensure that women do not miss their first cervical cancer screening when they become eligible. Therefore, implementing measures to encourage individuals to undergo their first cervical cancer screening is crucial. 

The coronavirus disease 2019 (COVID-19) pandemic has further reduced cervical cancer screening coverage in many countries, thus resulting in the implementation of alternative cervical cancer screening methods, such as HPV self-sampling. COVID-19 resulted in the deaths of many people worldwide, and a lockdown was implemented to prevent the spread of the disease [9]. This made medical care dissemination challenging, consequently worsening cancer prognosis [10]. Additionally, temporary interruptions in cervical cancer screening have been observed in several countries, resulting in low screening rates [11,12]. In Japan, the cervical cancer screening rate has also declined without the implementation of lockdown measures in 2020 [13]. 

Persistent HPV infection is a known cause of cervical cancer [14]. HPV testing has better sensitivity than cytology alone for detecting cervical intraepithelial neoplasia of grade 2 or worse grade (CIN2+) [14,15,16]. HPV self-sampling is a screening method where individuals collect a vaginal sample themselves and send it for HPV testing [17,18]. This method differs from physician-administered HPV tests. The concordance rate between self- and physician-collected HPV samples is 85–90%, indicating that HPV self-sampling is equally as sensitive as physician sampling for detecting CIN2+ lesions [17,19,20]. In Sweden, HPV self-sampling was adopted as an alternative to routine screening during the COVID-19 pandemic, leading to an increase in screening coverage from 54% in 2020 to 60% in 2021 [21]. Therefore, HPV self-sampling proves to be an effective screening method during a pandemic. However, no studies in Japan have specifically examined the acceptance or benefits of this approach for improving cervical cancer screening rates among young unscreened women. In this study, we aimed to assess the return rates of HPV self-sampling kits and the acceptance of this method among 24-year-old Japanese women who had never previously been screened for cervical cancer during the COVID-19 pandemic.

## 2. Materials and Methods

### 2.1. Cervical Cancer Screening in the Fukui Prefecture

In Japan, municipalities are primarily responsible for cervical cancer screening and cover almost all screening costs, including the costs associated with cervical cancer screening for pregnant women [22]. Publicly funded cervical cancer screening is performed using cytology [4]. 

Based on residence records, municipalities mail information on cervical cancer screenings to residents aged >20 years. Invitations for screening are mailed every 2 years to eligible women. Individuals who undergo screening are reported to the municipality that issued the invitation. Therefore, municipalities are aware of all women who have not been examined. The Fukui Prefecture offers cervical cancer screening at clinics, hospitals, and group screening sites. Pregnant women are screened for cervical cancer at hospitals and clinics. The number of women in Fukui Prefecture eligible for cervical cancer screening every two years, based on the 2022 national census, is presented in Appendix A. The Fukui Health Care Association receives all cervical cancer specimens, and cytology is performed by skilled cytologists and pathologists. The cytological results are mailed to the women who undergo screening. Screened women with a cytological diagnosis of atypical squamous cells of undetermined significance (ASCUS) or higher receive a notice recommending secondary screening at a hospital in the Fukui Prefecture. During secondary screening, women are managed according to the guidelines for obstetrics and gynecology. A colposcopic biopsy is performed if the cytology result is a low-grade squamous intraepithelial lesion (LSIL) or higher (4). A high-risk HPV test or a repeat cytology examination is performed for ASCUS. If the HPV test is positive, a colposcopic biopsy is performed. Performing a colposcopic biopsy is necessary if a repeat cytology shows ASCUS or higher (4). The Fukui Prefecture Health Care Association has reported the results of the cytological and histological examinations during secondary screening [23].

### 2.2. Cervical Cancer Statistics in the Fukui Prefecture During the COVID-19 Pandemic

COVID-19 spread rapidly from its origin in Wuhan, China, reaching pandemic status just a month after its initial discovery [24]. On 14 January 2020, Japan confirmed its first case of the infection [25]. Several countries closed their cities to prevent the spread of the disease and delay the peak of the pandemic [9]. On 7 April 2020, the government in Japan declared a state of emergency in Japan, allowing curfews and business closures [26]. The government requested citizens to adopt new hygiene and social practices to curb the spread of the disease, such as thorough handwashing and avoiding social gatherings. Subsequently, more restrictive and costly measures were introduced, including school closures and stay-at-home orders [26]. In contrast to the lockdowns enforced in countries like Italy, France, Germany, the United Kingdom, and the United States, the Japanese government’s declaration of a state of emergency did not include penalties for individuals who failed to adhere to these requests [26]. Prefectural authorities urged residents to avoid inter-prefectural travel, non-essential outings, and crowded places [27]. As a result, cervical cancer screenings continued in Fukui Prefecture during the COVID-19 pandemic. Data from the Fukui Prefecture Health Management Association were analyzed to assess the number of cervical cancer screenings conducted in the region from 2018 to 2020.

### 2.3. Study Population

Studies have previously examined a variety of unscreened populations, including women who have not been screened in more than 10 years in spite of repeated reminders [28,29]. This study included 24-year-old women without a history of cervical cancer screening who lived in Echizen City, Ono City, Katsuyama City, Sakai City, Sabae City, Fukui City, or Wakasa Town, Japan. In May 2021, seven municipalities extracted the data of eligible women for this study. A cytological examination was performed on the pregnant women during their pregnancy check-ups; therefore, they were excluded from this study. However, those who became pregnant after participating in this study were included. 

### 2.4. Opt-In HPV Self-Sampling

This study used an opt-in method for HPV self-sampling. Seven municipalities sent letters to all women aged 24 years who had never been screened for cervical cancer in August 2021. The letter included a QR code and a web address for registration. An eligibility assessment was conducted at the University of Fukui. Women who agreed to participate in this study received HPV self-sampling kits and questionnaires from the Fukui Prefecture Healthcare Association.

Self-sampling was performed using an Evalyn brush in this study. The use of the Evalyn brush in HPV self-sampling showed results similar to that observed in physician HPV sampling, and CIN2+ detection rates were comparable to those of physician HPV sampling [20,30,31]. HPV DNA testing results using Evalyn brushes were not affected by specimen storage conditions [32]. An instruction manual was included with the Evalyn brush kit prepared by the Japan Cancer Society. After self-sampling, an Evalyn brush kit and questionnaire were enclosed in an envelope by the participants. These were sent to the Fukui Prefecture Health Care Association. An HPV self-sampling study was conducted in four municipalities in the Fukui Prefecture without any reported issues [23]. This study used a similar scheme [23].

### 2.5. HPV Testing Using the Evalyn Brush

The Evalyn brush was used in conjunction with a Cobas 4800 polymerase chain reaction (PCR)-based HPV DNA test. The rate of CIN2+ detection is higher in PCR-based HPV DNA tests compared to that in signal-based HPV tests [33]. The results of the use of the Evalyn brush in combination with the Cobas 4800 system has a high concordance rate with the results of the HPV samples collected by physicians [20]. In this study, the Cobas 4800 was used to detect HPV-16 and -18, and 12 other high-risk HPV genotypes (HPV others), including HPV-31, -33, -35, -39, -45, -51, -52, -56, -58, -59, -66, and -68. All tests were conducted following the manufacturer’s instructions [16,20,34]. Previous clinical studies have demonstrated that the majority of HPV self-sampling kits returned by participants gave accurate results [23].

### 2.6. Management After HPV Self-Sampling

After HPV self-sampling, the participants were managed as previously described [23]. Participants received self-sampled HPV test results by mail. Cervical cancer screening was recommended for those who tested positive for HPV in the Fukui Prefecture. Despite being HPV-negative, these women were advised to undergo regular cervical cancer screening. Results of the cytological tests were mailed to participants. An ASCUS or higher cytological diagnosis required participants to visit a clinic or hospital in the Fukui Prefecture. Participants underwent secondary screening in accordance with the Japan Obstetrics and Gynecology Practice Guidelines 2020 [4].

### 2.7. Statistical Analyses

Frequencies and proportions were used to describe categorical variables. The EZR version 1.42 (Saitama Medical Center, Jichi Medical University, Saitama, Japan), a graphical user interface for R (The R Foundation for Statistical Computing, Vienna, Austria) [35], was used for all statistical analyses,. The results sent to the Fukui Health Care Association by March 2022 were analyzed in this study.

### 2.8. Ethics Approval

The Research Ethics Committee of the University of Fukui (protocol code: 20200014) approved this study on 30 April 2020. All participants in the study provided informed consent before taking part in the study.

## 3. Results

Table 1 shows the details of the screening participants for cervical cancer in the Fukui Prefecture from 2018 to 2020. The cervical cancer screenings performed on women in 2020 were 76.9% (20,101/26,128) of the value in 2018. The number of individuals of all ages screened declined in 2020 compared to that in 2018. From 2018 to 2020, the percentage of participants aged 20–24 years were 84.9% (745/877) of the value in 2018, whereas the percentage of participants aged 25–29 years were 80.0% (1203/1503) of the value in 2018. Additionally, the 24-year-olds who underwent their first screening in 2020 were 82.6% (114/138) of the value in 2018.

Figure 1 shows the details of the study participants. This study included 1997 individuals aged 24 years with no history of cervical cancer screening in Echizen City, Ono City, Katsuyama City, Sakai City, Sabae City, Fukui City, and Wakasa Town, which were in the Fukui Prefecture. Overall, 13.4% (268/1997) of the eligible participants agreed to participate in the study. A self-sampling kit for HPV was requested. In total, 72.4% (194/268) of participants returned the kits, resulting in a return rate of 9.7% (194/1997) from the eligible population. In total, 13.9% (27/194) of those who returned the kits underwent cytology after submission, and 11.1% (3/27) of them underwent cytology during the pregnancy checkup. Among the participants who did not return the kit, 9.5% (7/74) underwent cytology. In total, 12.7% (34/268) underwent cytology after providing consent for HPV self-sampling. 

Self-sampling HPV testing resulted in a 14.9% positivity rate (29/194). There were 41.4% (12/29) of HPV positive participants who underwent cytology. Cytology was performed on 15.1% (15/165) of the HPV negative participants. After obtaining consent for HPV self-sampling, the total number of cervical cancer screening participants consisted of participants who returned the kit and those who did not return the kit and underwent cytology [23]. After consent was obtained for HPV self-sampling, the overall cervical cancer screening rate was 75.0% (201/268). Excluding cytology at the time of the pregnancy checkup for participants who did not return the kit, the overall cervical cancer screening rate was 73.5% (197/268).

HPV-16 was detected in 3.4% (1/29) of the participants and HPV others in 96.6% (28/29). Some participants with negative in the HPV self-sampling test underwent a cytological examination, and the cytology results were negative for intraepithelial lesion or malignancy (NILM). Amongst the HPV-positive cases, 16.7% (2/12) had ASCUS, 8.3% (1/12) had LSIL, and 75% (9/12) had NILM. 

Table 2 presents the HPV self-sampling questionnaire results. According to the questionnaire, 57.8% (111/192) of the participants who returned the kit reported no pain during self-sampling. Regarding ease of use, of the 192 participants who returned the kit, 53.6% (103/192) found it to be easy to use, and the instructions were easy to understand for 78% (150/192) of the participants who returned the kit. In addition, it was found that 72.9% (140/192) of the participants who returned the kit preferred regular HPV self-sampling.

## 4. Discussion

This study surveyed 24-year-old women with no prior history of cervical cancer screening to evaluate the return rate of opt-in HPV self-sampling and its acceptability among women with no previous screening during the COVID-19 pandemic. Compared to 2018, the number of cervical cancer screenings dropped to 76.9% in 2020. From the seven municipalities, a total of 1997 eligible women were selected, with 13.4% agreeing to participate. Among the participants, 72.4% returned the HPV self-sampling kit, resulting in a return rate of 9.7% among the eligible women. Of those who returned the kits and tested positive for HPV, 42.4% underwent cytology. Furthermore, 72.9% of participants who returned the kits expressed a desire to continue with regular HPV self-sampling. Overall, the return rate for HPV self-sampling among women who had not been previously screened was favorable, and the acceptance of this method appeared promising.

Our study demonstrated that opt-in HPV self-sampling for 24-year-old women who had never previously undergone cervical cancer screening resulted in favorable kit return rates. Across several studies, among women who had not undergone cervical cancer screening for periods ranging from 3 to 10 years, return rates for opt-in HPV self-sampling kits varied from 63.2% to 79.5%, with eligible population return rates ranging from 8.15% to 20.7% [28,29,36]. Our previous study in Fukui Prefecture indicated that 77.3% of participants in their 30s without cervical cancer screening for more than 5 years returned their HPV self-sampling kits, with a return rate of 8.2% for the eligible women [23]. However, our study included women who had not experienced cervical cancer screening after the initiation of screening, which represents a different eligible population compared to that in other studies. Among women without screening in the 6 months following their first screening at age 24, the overall screening participation rate, including both HPV testing and cytology, was 16.2% of eligible population when opt-in HPV self-sampling was offered [37]. Our research specifically focused on a group who had not undergone cervical cancer screening for 5 years after reaching the recommended age for screening, and there are no previous studies that have been limited to this group. Our findings indicated that introducing HPV self-sampling to women who had never experienced cervical cancer screening long term after the recommended age for initiating cervical cancer screening, particularly during the COVID-19 pandemic, resulted in favorable return rates. The high return rates for HPV self-sampling reflect its acceptability among the targeted women in their 20s. Therefore, the implementation of HPV self-sampling could encourage screening uptake within this age group, potentially improving public health outcomes and reducing cervical cancer incidence and mortality in Japan. However, many women who had never undergone cervical cancer screening did not participate in HPV self-sampling. It is necessary to investigate the reasons for non-participation to improve the effectiveness of HPV self-sampling.

Many participants in our study expressed a willingness to continue using HPV self-sampling. Psychological barriers are often encountered when undergoing cervical cancer screening for the first time. In a survey of Japanese women aged 19–24 years who had never experienced cervical cancer screening, the most common reasons cited for not undergoing screening were discomfort with visiting an obstetrician and being too busy to make an appointment [38]. Compared to healthcare provider-conducted screening, HPV self-sampling is generally more acceptable [39], as it is perceived as easier and more comfortable [40]. Unscreened Japanese women favor HPV self-sampling due to reduced pain, less embarrassment, and a more comfortable experience [41]. Notably, our study is the first to demonstrate a favorable acceptance of HPV self-sampling among women who had never been screened before during the COVID-19 pandemic.

Cytology is infrequently performed following a positive HPV self-sampling test result. While self-sampling can promote cervical cancer screening among previously unscreened women and increase screening uptake compared to standard mailed invitations [42,43,44,45,46], only 40% of participants with HPV-positive results proceeded with cytology in this study. In contrast, a previous study found that 61.5% of HPV-positive women in their 30s who had not been screened before underwent cytology [23]. This difference may be due to the use of cytology as a confirmatory test for women who had never been screened. Additionally, the clinical implications of HPV positivity in Japan may not be widely understood [47], highlighting an area for future research.

Self-sampling HPV testing resulted in a 14.9% positivity rate in this study. A previous study in Fukui prefecture reported an HPV test positivity rate of 16.5% for the 25- to 29-year-old age group [16]. In this study, most of the detected HPV types were categorized as HPV others including HPV-31, -33, -35, -39, -45, -51, -52, -56, -58, -59, -66, and -68. This study was unable to specify the individual HPV types within the others category. Previous studies have indicated that the most frequently detected HPV type among Japanese women aged 20 to 25 was HPV-52 (8.1%), followed by HPV-16 (6.5%), HPV-51 (4.5%), HPV-18 (4.0%), and HPV-31 (3.8%) [48]. The HPV positivity rate in our study was similar to that reported in a previous study [16]. The detected HPV types differed from those identified in another previous study [48]. It is important to note that this study only included 24-year-old women who had never been screened, which differs from the general population. Therefore, it may be necessary to analyze all high-risk HPV types to accurately assess the risk of cervical cancer.

This study had some limitations. The participants were 24-year-old women and other age group women were not examined. The purpose of this study was to assess the impact of implementing HPV self-sampling within the context of the cervical cancer screening framework. As a result, we were not able to compare opt-in HPV self-sampling with opt-out methods or reminder mailings [23]. To generalize and understand the impact of opt-in HPV self-sampling on short- and long-term health outcomes, larger multicenter studies are needed. As this study was carried out during the COVID-19 pandemic period, studies need to be carried out in the post-COVID-19 periods. 

## 5. Conclusions

In conclusion, the acceptability and kit return rates for HPV self-sampling are good. However, the follow-up rate after HPV self-sampling is low, and further studies are required to improve this rate. 

## Figures and Tables

**Figure 1 healthcare-12-02160-f001:**
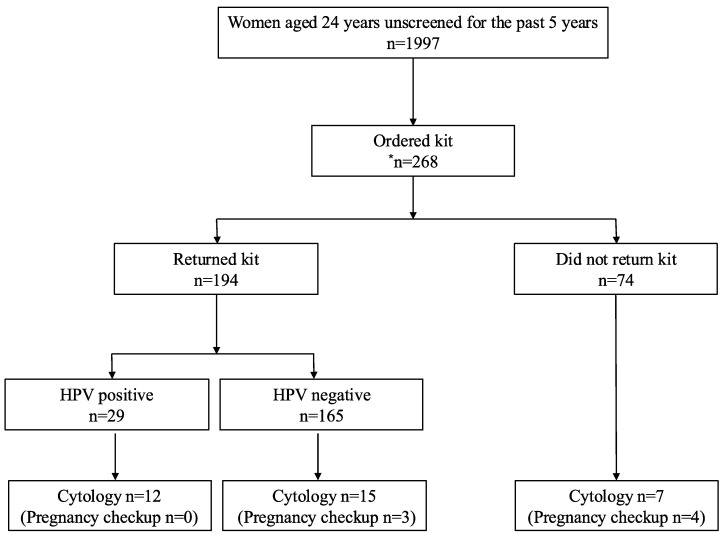
Study flowchart. * Initially, 282 participants agreed to participate; however, 14 withdrew their consent. HPV, human papillomavirus.

**Table 1 healthcare-12-02160-t001:** Cervical cancer screening from 2018 to 2020.

Age (Years)	2018	2019	2020
All ages (n)	26,128	25,612	20,101
20–24 (n)	877	814	745
25–29 (n)	1503	1425	1203
30–34 (n)	2166	2009	1603
35–39 (n)	2671	2299	2079
40–44 (n)	3097	2796	2356
45–49 (n)	2723	2833	2335
50–54 (n)	2274	2464	1895
55–59 (n)	2153	2234	1660
60–64 (n)	2273	2300	1743
>65 (n)	6391	6438	4482
First cervical cancer screening			
20 (n)	37	37	30
24 (n)	138	127	114

Note: n = total number of women screened for cervical cancer.

**Table 2 healthcare-12-02160-t002:** Participants’ opinions on HPV self-sampling kit use.

Pain	n	%
Strong	10	5.2
Moderate	68	35.4
Nothing	111	57.8
No response	3	1.6
**Usability**		
Difficult	15	7.8
Intermediate	72	37.5
Easy	103	53.6
No response	2	1.0
**Ease of understanding instructions**		
Difficult	5	2.6
Intermediate	36	18.8
Easy	150	78.1
No response	1	0.5
**Undergo testing on a regular basis**		
Yes	140	72.9
No	4	2.1
No idea	48	25.0
No response	0	0.0

Note: Two participants did not return the questionnaire. n = total number of women who returned the kit and answered the questionnaire. % = percentage of women who returned the kit and indicated an impression.

## Data Availability

The data presented in this study are available from the corresponding author upon request.

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
