# Peer review of "Human Papillomavirus Self-Sampling for Unscreened Women Aged 24 Years During the COVID-19 Pandemic"

_healthcare, 2024, doi:10.3390/healthcare12212160_

Round 1
Reviewer 1 Report
Comments and Suggestions for Authors
The paper describes the applicability of and responsiveness to self-sampling in previously unscreened women of the Fukui Prefecture. Overall, the study is well-designed and well-written. The results are relevant for public health and can be applied for future healthcare guidelines. My only note is that the flowchart (Figure 1) is blank.
Author Response
Reviewer 1:
Reviewer comments 1:
The paper describes the applicability of and responsiveness to self-sampling in previously unscreened women of the Fukui Prefecture. Overall, the study is well-designed and well-written. The results are relevant for public health and can be applied for future healthcare guidelines. My only note is that the flowchart (Figure 1) is blank.
Response 1:
We would like to express our sincere gratitude for your time and valuable feedback on our manuscript. Your thoughtful comments have significantly contributed to the improvement of our work, and we truly appreciate the effort you invested in reviewing our manuscript. We have revised our paper based on your suggestions.
The uploaded Word file contains the study flowchart. The previous conversion from the Word file to a PDF did not work as expected. We have replaced the study flowchart in the Word file with a new version.
Reviewer 2 Report
Comments and Suggestions for Authors
In the submitted manuscript “Human Papillomavirus Self-Sampling for Unscreened Women Aged 24 Years During the COVID-19 Pandemic”, this study presents opt-in HPV self-sampling among 24-year-old women who had never been screened for cervical cancer had a favorable kit return rate and was well accepted by the participants. However, there are some points that need to be clarified:
1. Figure 1, study flowchart disappears, please upload it.
2. The authors showed that HPV-16 was detected in 3.4% (1/29) of the participants and other HPV genotypes in 96.6% (28/29). The authors should mention the prevalence (or percentage) of each genotype that were found in the study population.
3. Line 224, please correct the word to “All HPV-positive cases”.
4. This study showed that self-sampling HPV testing resulted in a 14.9% positivity rate (29/194). Is it higher or lower than previous reports or other studies? The authors should briefly discuss this point.
5. Why the follow-up rate after HPV self-sampling is low? Only 40% of participants with HPV-positive results proceeded with cytology in this study. What is the reason behind this? The authors should discuss this point.
Author Response
Reviewer 2:
Reviewer comments:
In the submitted manuscript “Human Papillomavirus Self-Sampling for Unscreened Women Aged 24 Years During the COVID-19 Pandemic”, this study presents opt-in HPV self-sampling among 24-year-old women who had never been screened for cervical cancer had a favorable kit return rate and was well accepted by the participants. However, there are some points that need to be clarified:
Response:
We would like to express our sincere gratitude for your time and valuable feedback on our manuscript. Your thoughtful comments have significantly contributed to the improvement of our work, and we truly appreciate the effort you invested in reviewing our manuscript. We have revised our paper based on your suggestions.
Reviewer comments 1:
1. Figure 1, study flowchart disappears, please upload it.
Response 1:
The uploaded Word file contains the study flowchart. The previous conversion from the Word file to a PDF did not work as expected. We have replaced the study flowchart in the Word file with a new version.
Reviewer comments 2:
The authors showed that HPV-16 was detected in 3.4% (1/29) of the participants and other HPV genotypes in 96.6% (28/29). The authors should mention the prevalence (or percentage) of each genotype that were found in the study population.
Response 2:
The Cobas 4800 system was employed to detect HPV-16, HPV-18, and an additional 12 high-risk HPV genotypes (HPV others), including HPV-31, -33, -35, -39, -45, -51, -52, -56, -58, -59, -66, and -68. However, the Cobas 4800 cannot identify each HPV type individually; instead, these are grouped under the "Others" category. Consequently, this study was unable to specify the individual HPV types included within the "Others" category. Previous studies have shown that the most frequently detected HPV type in Japanese women aged 20 to 25 was HPV-52 (8.1%), followed by HPV-16 (6.5%), HPV-51 (4.5%), HPV-18 (4.0%), and HPV-31 (3.8%) (Konno et al. 2011). This study only included 24-year-old women who had been never screened. It differs from the normal population. It may be necessary to analyze all high-risk HPV types in order to assess the risk of cervical cancer.
Line 168
In this study, the Cobas 4800 was used to detect HPV-16 and -18, and 12 other high-risk HPV genotypes (HPV others), including HPV-31, -33, -35, -39, -45, -51, -52, -56, -58, -59, -66, and -68. All tests were conducted following the manufacturer's instructions [18,20,34].
Line 228
HPV-16 was detected in 3.4% (1/29) of the participants and HPV others in 96.6% (28/29)
Line 302
Self-sampling HPV testing resulted in a 14.9% positivity rate in this study. A previous study in Fukui prefecture reported an HPV test positivity rate of 16.5% for the 25 to 29-year-old age group [18]. In this study, most of the detected HPV types were categorized as HPV others including HPV-31, -33, -35, -39, -45, -51, -52, -56, -58, -59, -66, and -68. This study was unable to specify the individual HPV types within the others category. Previous studies have indicated that the most frequently detected HPV type among Japanese women aged 20 to 25 was HPV-52 (8.1%), followed by HPV-16 (6.5%), HPV-51 (4.5%), HPV-18 (4.0%), and HPV-31 (3.8%) [48]. The HPV positivity rate in our study was similar to that reported in the previous study [18]. The detected HPV types differed from those identified in previous study [48]. It is important to note that this study only included 24-year-old women who had never been screened, which differs from the general population. Therefore, it may be necessary to analyze all high-risk HPV types to accurately assess the risk of cervical cancer.
Reviewer comments 3:
3. Line 224, please correct the word to “All HPV-positive cases”.
Response 3:
Thank you for pointing this out. In this study, some participants who tested negative in the HPV self-sampling underwent a cytological examination. Cytology result is NILM
Line 229
Some participants with negative in the HPV self-sampling test underwent a cytological examination, and the cytology results were negative for intraepithelial lesion or malignancy (NILM).
Reviewer comments 4:
4. This study showed that self-sampling HPV testing resulted in a 14.9% positivity rate (29/194). Is it higher or lower than previous reports or other studies? The authors should briefly discuss this point.
Response 4:
Self-sampling HPV testing resulted in a 14.9% positivity rate in this study. A previous study in Fukui prefecture reported an HPV test positivity rate of 16.5% for the 25 to 29-year-old age group (Kurokawa et al. 2018)
Line 302
Self-sampling HPV testing resulted in a 14.9% positivity rate in this study. A pre-vious study in Fukui prefecture reported an HPV test positivity rate of 16.5% for the 25 to 29-year-old age group [18]. In this study, most of the detected HPV types were categorized as HPV others including HPV-31, -33, -35, -39, -45, -51, -52, -56, -58, -59, -66, and -68. This study was unable to specify the individual HPV types within the others category. Pre-vious studies have indicated that the most frequently detected HPV type among Japanese women aged 20 to 25 was HPV-52 (8.1%), followed by HPV-16 (6.5%), HPV-51 (4.5%), HPV-18 (4.0%), and HPV-31 (3.8%) [48]. The HPV positivity rate in our study was similar to that reported in the previous study [18]. The detected HPV types differed from those identified in previous study [48]. It is important to note that this study only included 24-year-old women who had never been screened, which differs from the general population. Therefore, it may be necessary to analyze all high-risk HPV types to accurately assess the risk of cervical cancer.
Reviewer comments 5:
5. Why the follow-up rate after HPV self-sampling is low? Only 40% of participants with HPV-positive results proceeded with cytology in this study. What is the reason behind this? The authors should discuss this point
Response 5:
Thank you for your suggestion. This may be attributed to the fact that the study population had not previously undergone cytological examinations. Furthermore, it is possible that the implications of HPV positivity were not fully understood by the study participants.
Line 298
This difference may be due to the use of cytology as a confirmatory test for women who had never been screened. Additionally, the clinical implications of HPV positivity in Japan may not be widely understood [47], highlighting an area for future research.
Reviewer 3 Report
Comments and Suggestions for Authors
This is an informative article about HPV self-sampling in Japan, and will contribute to the evidence base about the acceptability of HPV self-sampling around the world.
Introduction: Another challenge is that HPV infections in this age group are often transient, and common as well.
I’m wondering if you should briefly bring up the challenges around HPV vaccination in Japan? Because so many women are unvaccinated, this points to the greater need for on-time cervical cancer screening.
Methods:
Page 3: line 143: Were these letters sent to all women aged 24 years who were never screened, or was any type of population-based sampling done prior to contacting eligible women?
Figure 1: The study flowchart shows up as blank for me.
Table 1: Are denominator data available so that the percentage of women screened by each age group could be presented?
Lines 190 – 196: The findings are written as if they are expressed as a percentage change, but the calculations don’t seem to align with that formula. For example, the sentence “the cervical cancer screenings performed on women in 2020 decreased by 76.9% (20,101/26,128) compared with that in 2018” may need to be written as “the cervical cancer screenings performed on women in 2020 were 76.9% of the value in 2018, or 23% lower.)”
Line 223: I’m curious about the other HPV genotypes detected. Are you able to provide more details as to which ones and how common they were?
Lines 268-269: This seems to be a bit of an overstatement because a significant number of women did not initially opt in for screening. Although self-sampling seems very promising in this population, it would be interesting to follow up with women who weren’t interested in self-sampling to understand why, which could inform future outreach efforts.
Comments on the Quality of English LanguageOnly minor edits are needed. The quality of the English language was fine overall.
Author Response
Reviewer 3:
Reviewer comments:
This is an informative article about HPV self-sampling in Japan, and will contribute to the evidence base about the acceptability of HPV self-sampling around the world.
Response:
We would like to express our sincere gratitude for your time and valuable feedback on our manuscript. Your thoughtful comments have significantly contributed to the improvement of our work, and we truly appreciate the effort you invested in reviewing our manuscript. We have revised our paper based on your suggestions.
Reviewer comments 1:
Introduction: Another challenge is that HPV infections in this age group are often transient, and common as well.
I’m wondering if you should briefly bring up the challenges around HPV vaccination in Japan? Because so many women are unvaccinated, this points to the greater need for on-time cervical cancer screening.
Response 1:
Thank you for the excellent suggestion. We agree with your point. We recognize that the low HPV vaccination rate is a significant issue in Japan. Therefore, we believe it is crucial to increase the cervical cancer screening rate. We have incorporated this perspective in the Introduction section.
Line 36
Women in Japan in their 20s have a high incidence of cervical cancer. In 2013, the incidence rate was 6.20 per 100,000 population, as reported by the Global Cancer Observatory. In comparison, the United States and Australia reported cervical cancer incidence of 3.18 and 5.59 per 100,000 population, respectively [1]. In 2022, the incidence among Japanese women in their twenties had increased to 9.90 per 100,000 population, whereas the incidence rates in the United States and Australia were 4.31 and 3.17 per 100,000 population, respectively [1]. These findings underscore the high and escalating incidence and mortality rates of cervical cancer in Japan, contrasting with the trends observed in majority of developed countries [2]. The significant factors contributing to this trend are the low HPV vaccination rate and the low acceptance of cervical cancer screening [2]. Despite the resumption of government recommendations, HPV vaccination coverage remains low, with only 2.83% of the vaccine-resumed generation (birth fiscal year 2010) receiving the vaccine [3]. For this reason, it is essential to increase the cervical cancer screening rate.
Reviewer comments 2:
Methods:
Page 3: line 143: Were these letters sent to all women aged 24 years who were never screened, or was any type of population-based sampling done prior to contacting eligible women?
Response 2:
Based on residence records, municipalities mail information on cervical cancer screenings to residents aged >20 years. Individuals who undergo screening are reported to the municipality that issued the invitation. Therefore, municipalities are aware of all women who have not been examined. In this study, seven municipalities sent letters to all women aged 24 years who had never been screened for cervical cancer.
Line 147
Seven municipalities sent letters to all women aged 24 years who had never been screened for cervical cancer in August 2021.
Reviewer comments 3:
Figure 1: The study flowchart shows up as blank for me.
Response 3:
The uploaded Word file contains the study flowchart. The previous conversion from the Word file to a PDF did not work as expected. We have replaced the study flowchart in the Word file with a new version.
Reviewer comments 4:
Table 1: Are denominator data available so that the percentage of women screened by each age group could be presented?
Response 4:
Table 1 presents the data of cervical cancer screening analyzed by the Fukui Prefecture Health Management Association. The exact number of individuals invited for cervical cancer screening in Fukui Prefecture for the years 2018, 2019, and 2020 is unavailable. Women aged 20 years or older are advised to undergo cytological screening for cervical cancer every two years in Japan. As supplementary information, we have included data on the population of Fukui Prefecture based on the national census in 2022, indicating that half of the women in this population are eligible for cervical cancer screening.
Line 105
The number of women in Fukui Prefecture eligible for cervical cancer screening every two years, based on the 2022 national census, is presented in Supplementary Table 1.
Reviewer comments 5:
Lines 190 – 196: The findings are written as if they are expressed as a percentage change, but the calculations don’t seem to align with that formula. For example, the sentence “the cervical cancer screenings performed on women in 2020 decreased by 76.9% (20,101/26,128) compared with that in 2018” may need to be written as “the cervical cancer screenings performed on women in 2020 were 76.9% of the value in 2018, or 23% lower.)”
Response 5:
Thank you for your suggestion. We made the following changes.
Line 194
Table 1 shows the details of the screening participants for cervical cancer in the Fukui Prefecture from 2018 to 2020. The cervical cancer screenings performed on women in 2020 were 76.9% (20,101/26,128) of the value in 2018. The number of individuals of all ages screened declined in 2020 compared to that in 2018. From 2018 to 2020, the percentage of participants aged 20–24 years were 84.9% (745/877) of the value in 2018, whereas the percentage of participants aged 25–29 years were 80.0% (1,203/1,503) of the value in 2018. Additionally, 24-year-olds underwent their first screening in 2020 were 82.6% (114/138) of the value in 2018.
Reviewer comments 6:
Line 223: I’m curious about the other HPV genotypes detected. Are you able to provide more details as to which ones and how common they were?
Response 6:
The Cobas 4800 system was employed to detect HPV-16, HPV-18, and an additional 12 high-risk HPV genotypes, including HPV-31, -33, -35, -39, -45, -51, -52, -56, -58, -59, -66, and -68. However, the Cobas 4800 cannot identify each HPV type individually; instead, these are grouped under the "Others" category. Consequently, this study was unable to specify the individual HPV types included within the "Others" category. Previous research has indicated that, among Japanese women aged 20 to 25, the most frequently detected HPV type was HPV-52 (8.1%), followed by HPV-16 (6.5%), HPV-51 (4.5%), HPV-18 (4.0%), and HPV-31 (3.8%) (Konno et al. 2011).
Line 168
In this study, the Cobas 4800 was used to detect HPV-16 and -18, and 12 other high-risk HPV genotypes (HPV others), including HPV-31, -33, -35, -39, -45, -51, -52, -56, -58, -59, -66, and -68. All tests were conducted following the manufacturer's instructions [18,20,34].
Line 304
In this study, most of the detected HPV types were categorized as HPV others including HPV-31, -33, -35, -39, -45, -51, -52, -56, -58, -59, -66, and -68. This study was unable to specify the individual HPV types within the others category. Previous studies have indicated that the most frequently detected HPV type among Japanese women aged 20 to 25 was HPV-52 (8.1%), followed by HPV-16 (6.5%), HPV-51 (4.5%), HPV-18 (4.0%), and HPV-31 (3.8%) [48]. The HPV positivity rate in our study was similar to that reported in the previous study [18]. The detected HPV types differed from those identified in previous study [48]. It is important to note that this study only included 24-year-old women who had never been screened, which differs from the general population. Therefore, it may be necessary to analyze all high-risk HPV types to accurately assess the risk of cervical cancer.
Reviewer comments 7:
Lines 268-269: This seems to be a bit of an overstatement because a significant number of women did not initially opt in for screening. Although self-sampling seems very promising in this population, it would be interesting to follow up with women who weren’t interested in self-sampling to understand why, which could inform future outreach efforts.
Response 7:
Thank you for your valuable suggestions. We agree with your recommendation. We believe that examining the factors contributing to the low uptake of HPV self-sampling will provide insights that can enhance the implementation and effectiveness of this method.
Line 278
However, many women who had never undergone cervical cancer screening did not participate in HPV self-sampling. It is necessary to investigate the reasons for non-participation to improve the effectiveness of HPV self-sampling.